# Half-Heusler-like compounds with wide continuous compositions and tunable p- to n-type semiconducting thermoelectrics

Zirui Dong [1], Jun Luo [1,2✉], Chenyang Wang[1], Ying Jiang[2], Shihua Tan [3], Yubo Zhang[3,4], Yuri Grin[5], Zhiyang Yu[6], Kai Guo [1], Jiye Zhang[1] & Wenqing Zhang [3,4✉]

Half-Heusler and full-Heusler compounds were considered as independent phases with a natural composition gap. Here we report the discovery of $TiRu_{1+x}Sb$ ($x = 0.15 \sim 1.0$) solid solution with wide homogeneity range and tunable p- to n-type semiconducting thermo-electrics, which bridges the composition gap between half- and full-Heusler phases. At the high-Ru end, strange glass-like thermal transport behavior with unusually low lattice thermal conductivity (~1.65 $Wm^{-1}K^{-1}$ at 340 K) is observed for $TiRu_{1.8}Sb$, being the lowest among reported half-Heusler phases. In the composition range of $0.15 < x < 0.50$, $TiRu_{1+x}Sb$ shows abnormal semiconducting behaviors because tunning Ru composition results in band structure change and carrier-type variation simultaneously, which seemingly correlates with the localized $d$ electrons. This work reveals the possibility of designing fascinating half-Heusler-like materials by manipulating the tetrahedral site occupancy, and also demonstrates the potential of tuning crystal and electronic structures simultaneously to realize intriguing physical properties.

[1] School of Materials Science and Engineering, Shanghai University, Shanghai 200444, China. [2] Materials Genome Institute, Shanghai University, Shanghai 200444, China. [3] Department of Physics and Shenzhen Institute for Quantum Science and Engineering, Southern University of Science and Technology, Shenzhen 518055, China. [4] Guangdong Provincial Key Lab for Computational Science and Materials Design, and Shenzhen Municipal Key-Lab for Advanced Quantum Materials and Devices, Southern University of Science and Technology, Shenzhen 518055, China. [5] Max-Planck-Institut für Chemische Physik fester Stoffe, Nöthnitzer Straße 40, 01187 Dresden, Germany. [6] State Key Laboratory of Photocatalysis on Energy and Environment, College of Chemistry, Fuzhou University, Fuzhou 350002, China. ✉email: junluo@shu.edu.cn; zhangwq@sustech.edu.cn

Since the first Heusler compound MnCu₂Al was discovered by Fritz Heusler in 1903[1], Heusler-based compounds have attracted persistent efforts due to their versatile and interesting properties[2]. Heusler compounds include half-Heusler (HH) and full-Heusler (FH) semiconductors/metals with the composition represented respectively as XYZ and XY₂Z (X and Y are metallic elements with different charges, while Z is the main group element served as the anion). The most electropositive X and the most electronegative Z form the face-centered-cubic (FCC) lattice in which the eight tetrahedral voids (interstitial sites) are either half or fully occupied. HH and FH compounds are thus defined according to the occupancy of the tetrahedral interstitial site, resulting in a composition gap between HH and FH. In both chemistry and materials science communities, the HH and FH compounds have been considered as independent phases with a natural HH-FH composition gap for nearly 100 years.

Semiconducting Heusler compounds are especially attractive for their great potential in thermoelectrics, spintronics, solar cells, diluted magnetic semiconductors, topological insulators, etc.[3]. Constrained by the valence electron rule[4,5], up to now only ~250 Heusler semiconductors have been discovered despite the relentless efforts on exploring Heusler semiconductors[1]. Despite the simple relationship of the crystal structures, in most ternary systems, the HH and FH phases are clearly separated by a two-phase region. Only a few Heusler-based phases with the compositions between HH and FH have been reported, such as TiFe₁.₃₃Sb[6], ZrNi₁₊δSn ($\delta = 0$–$0.05$)[7], and MCo₁.₅Sn (M = Ti, Zr, or Hf)[8]. These substances crystallize in the cubic non-centrosymmetric structure similar to Heusler compounds, but the occupancy of the tetrahedral interstitial site is between those of the HH and FH structures. TiFe₁.₃₃Sb and ZrNi₁₊δSn show a semiconducting behavior with promising thermoelectric properties while MCo₁.₅Sn (M = Ti, Zr, or Hf) compounds are ferromagnetic metals[6–8]. The discovery of the above-mentioned XY₁₊δZ materials indicates that the HH-like compounds with the compositions between HH and FH compounds are possible. Thus, it is greatly desired to bridge the composition gap between HH and FH compounds, which will expand the scope of Heusler-based compounds, especially with the semiconducting behavior. In addition, the appropriate choice of the components may also reduce the thermal conductivity in comparison with those of TiFe₁.₃₃Sb[6], ZrNi₁₊δSn[7], and MCo₁.₅Sn[8].

In this work, we report on the discovery of the TiRu₁₊ₓSb solid solution falling in the HH-FH composition gap and at the same time with tunable p- to n-type semiconducting thermoelectrics. We on purpose select the composition TiRuSb with 17 valence electrons[9] as the base to construct TiRu₁₊ₓSb ($x = 0$–$1$) materials, with the potential of filling the vacant tetrahedral interstitial site with Ru. To our surprise, the system shows a continuous variation of Ru composition covering a wide range of $0.15 < x < 1.0$, which keeps the structure of the HH phase. According to our experiments, both p- and n-type semiconductors are realized in the TiRu₁₊ₓSb solid solution by regulating the Ru content, and the sample with the composition around $x = 0.3$ behaves as an intrinsic semiconductor. Theoretical calculations confirm that the electronic band structure of TiRu₁₊ₓSb varies with the Ru content, resulting in the semiconducting behavior.

Figure 1a shows the crystal structure of HH and FH compounds, in which the most electropositive X, the most electronegative Z, and Y occupy the 4a, 4b, and tetrahedral interstitial sites of the FCC lattice, respectively. In the Heusler compounds adopting the $F\bar{4}3m$ space group, the tetrahedral interstitial sites are divided into 4c and 4d positions (solid blue and hollow spheres in Fig. 1a, respectively), and only half of them (4c) are occupied in HH phases. For FH compounds, all the tetrahedral

interstitial positions are occupied, and the space group changes to $Fm\bar{3}m$ because all the tetrahedral interstitial sites become indistinguishable (8c position). The synchrotron radiation X-ray powder diffraction (SR-XRD) patterns shown in Fig. 1b reveal that the TiRu₁₊ₓSb ($x = 0.1$–$1.0$) samples crystallize in the typical FCC lattice. The TiRuSb sample ($x = 0$) is a multi-phase one with the main HH phase (Supplementary Fig. S1). For the samples with $x = 0.1$, $0.2$, $0.7$, $0.8$, $0.9$, and $1.0$, only trace amounts of RuSb₂ as impurity are observed. No diffraction peaks from impurity phases are detected for the samples with $x = 0.3$–$0.6$. The diffraction peak (220) in all XRD patterns shifts continuously toward a lower $2\theta$ angle indicating that the unit cell expands monotonously with the increasing Ru content, which is confirmed by the change of lattice parameter shown in Fig. 1c. Except for the sample with $x = 0.1$, the refined lattice parameter $a$ increases monotonically with the Ru content, indicating that the Ru atoms are continuously filled into the vacant sites. The real sample compositions are very close to their nominal compositions, agreeing well with our chemical composition analysis by the electron probe micro-analysis (EPMA, see Supplementary Table S1). Based on the Ru content-dependent lattice parameter shown in Fig. 1c, the TiRu₁₊ₓSb solid solution has a homogeneity range of $0.15 < x \leq 1.0$.

The crystal structure of the single-phase TiRu₁.₅Sb sample has been refined from the SR-XRD data by the Rietveld method. Various structural models, such as partial ordering or random occupation of the tetrahedral interstitial sites, mixed occupation of some crystal sites, and so on, have been attempted. Careful refinement in the space group of $F\bar{4}3m$ for TiRu₁.₅Sb reveals the occupation of four Ti atoms at the 4a site, slightly off-center Sb position and partial occupation of two expected Ru positions (Ru shows only slight preference to occupy the 4c site). Thus, the TiRu₁.₅Sb has an HH-like crystal structure because it adopts the space group $F\bar{4}3m$ of a HH structure, but not the $Fm\bar{3}m$ space group of an FH compound. We name this structure as an HH-like one owing to the composition deviation and partial occupation of both 4c and 4d sites. According to the refinement result, the total number of Ru atoms per unit cell is 6.1 (3.2 Ru at 4c and 2.9 Ru at 4d), which agrees well with the nominal composition. The Sb atoms in TiRu₁.₅Sb are located at the 16e ($y$, $y$, 0.5-$y$) site with $y = 0.0119$, showing a small displacement from the original 4b position. Similar off-center location of atoms has also been reported in ScNiSb[10], TaGeIr[11], and HoPtSb[12] HH compounds.

The atomic structure of the TiRu₁.₅Sb sample is further analyzed by transmission electron microscopy (TEM). A typical dark field image and its corresponding energy dispersive spectroscopy (EDS) maps in Fig. 2a demonstrate a regular distribution of elements in the sample, with grain sizes ranging from 100 to 500 nm. The details of crystal structure have been clarified by further investigations at the atomic scale. Figure 2b shows an atom-resolved high-angle annular dark-field (HAADF) image and its corresponding atomic-scale EDS maps projected along a [100] direction, which indicates that Ti and Sb atoms locate at the 4a and 16e sites of the $F\bar{4}3m$ structure, respectively, while Ru atoms occupy the 4c and/or 4d site. In order to identify the three-dimensional occupation site of Ru atoms, the projection of Ru columns has been conducted along another direction ([110], Fig. 2c). According to the HH structure, each column in this projection is composed of a single-type lattice site, as indicated in Fig. 2c by the red, green, blue, and hollow blue spheres for the 4a, 16e, 4c, and 4d sites, respectively. As confirmed by the asymmetric HAADF line profile in Fig. 2d, both 4c and 4d sites are partially occupied by Ru atoms, whilst the occupancy of 4c sites is higher than that of 4d sites, agreeing well with the refinement results of the structure determination. Furthermore, the ratio of the 4c and 4d site occupancy factors varies depending on the

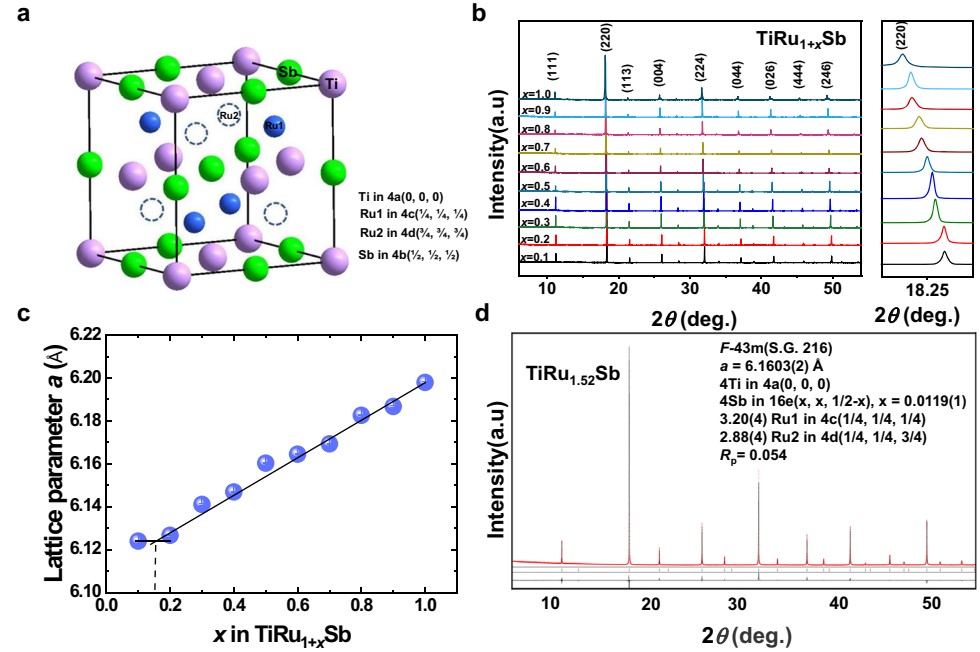

**Fig. 1 Crystal structure of TiRu$_{1+x}$Sb. a** Schematic atomic arrangement in Heusler-type compounds. **b** SR-XRD patterns of TiRu$_{1+x}$Sb samples. **c** Ru content-dependent lattice parameters of TiRu$_{1+x}$Sb samples. **d** SR-XRD pattern of the TiRu$_{1.5}$Sb sample together with the results of the crystal structure refinement.

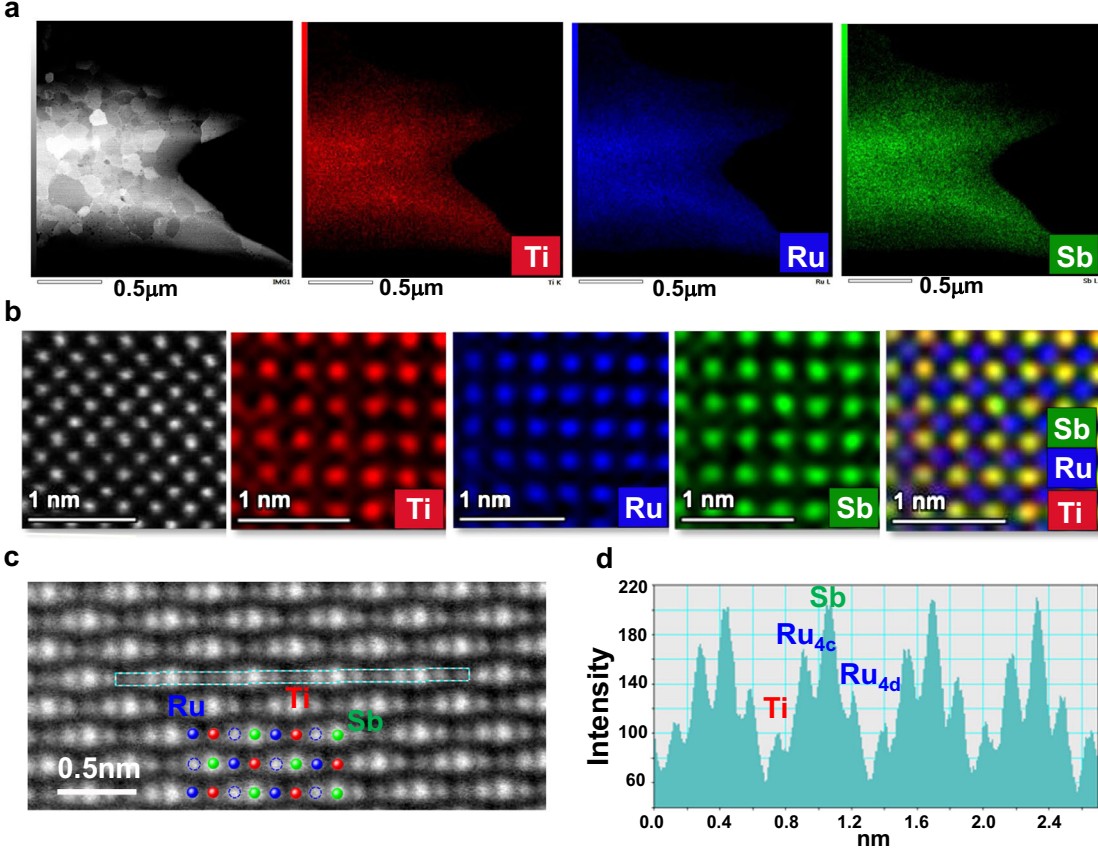

**Fig. 2 Microscopic structures of TiRu$_{1.5}$Sb. a** Dark field STEM image and corresponding EDS maps. **b** Atom-resolved HAADF image and corresponding EDS maps projected along the [100] direction. **c** Aberration correction high-resolution HAADF image along a [110] direction. **d** Intensity profile across the horizontal line in (**c**).

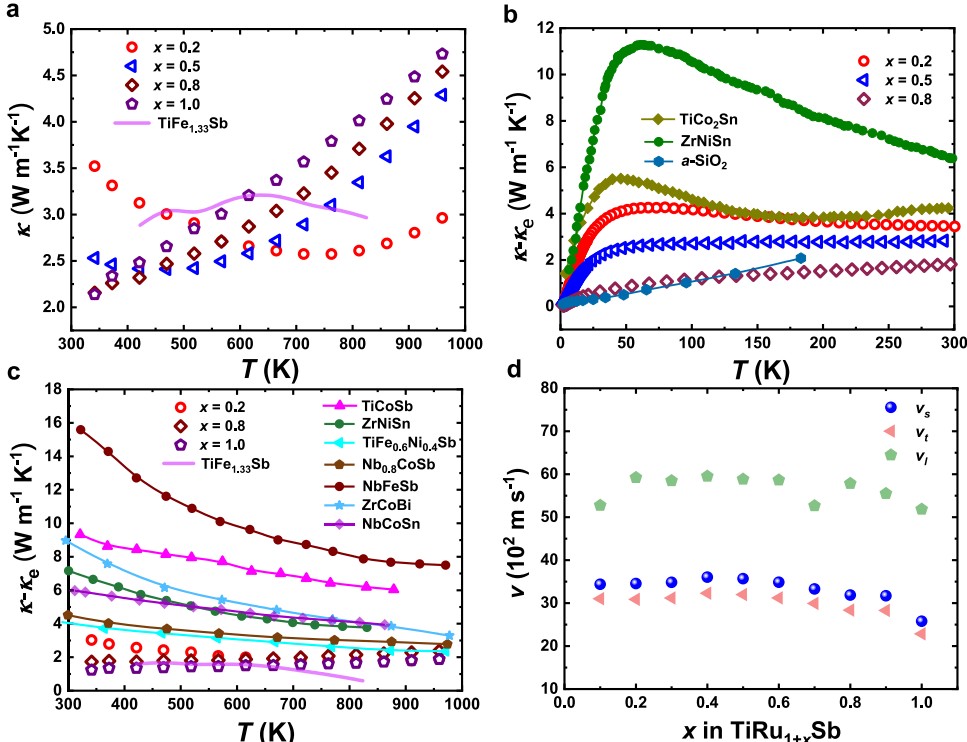

**Fig. 3 Thermal transport properties of TiRu$_{1+x}$Sb samples. a** Total thermal conductivity ($\kappa$) above room temperature. **b** Lattice thermal conductivity ($\kappa - \kappa_e$) from 2 to 300 K. **c** Lattice thermal conductivity ($\kappa - \kappa_e$) in comparison with characteristic HH and FH compounds in the temperature range of 300–1000 K. **d** Ru content-dependent average ($v_s$), longitudinal ($v_l$), and transverse ($v_t$) sound velocities. The solid lines in **a**–**c** are the experimental data of TiFe$_{1.33}$Sb, $\alpha$-SiO$_2$, ZrNiSn, TiCo$_2$Sn, and other typical HH compounds taken from literatures[6,15-22].

position of the sample, indicating local breaking of translational symmetry. This leads to the local changes in multi-center chemical bonding[13], which in turn may influence the lattice thermal conductivity[14].

The results of XRD and TEM analysis clearly prove that the HH-FH composition gap can be bridged by filling the vacant tetrahedral interstitial sites with appropriate atoms, with the Ru-based TiRu$_{1+x}$Sb solid solution as a representative and unambiguous example. Our work not only expands the Heusler family but also offers the opportunity to manipulate their physical properties by controlling the occupancy of tetrahedral interstitial sites. In the filled HH structure of TiRu$_{1.5}$Sb, the partial occupation of Ru atoms on both 4c and 4d sites should contribute to additional phonon scattering, leading to reduced lattice thermal conductivity. As shown in Fig. 3a, upon the filing of Ru, the total thermal conductivity $\kappa$ of the TiRu$_{1+x}$Sb sample at 340 K decreases dramatically from ~3.5 W m$^{-1}$ K$^{-1}$ for the sample with $x = 0.2$ to ~2.1 W m$^{-1}$ K$^{-1}$ for the sample with $x = 0.8$ (see Supplementary Fig. S2 for the thermal conductivities of all TiRu$_{1+x}$Sb samples), which are much lower than the typical HH semiconductors with total thermal conductivities normally up to 10 W m$^{-1}$ K$^{-1}$ and higher[15–21]. This unusually low $\kappa$ is greatly desired for thermoelectric HH compounds, and our work demonstrates the potential to significantly reduce $\kappa$ by filling the vacant sites.

To reveal the effect of filling Ru on the thermal transport properties, low-temperature thermal conductivity has been measured (see Supplementary Fig. S3 for the low-temperature transport properties of TiRu$_{1+x}$Sb samples). By subtracting the electronic contribution $\kappa_e$ from the total thermal conductivity (see Supplementary Section 1), the lattice thermal conductivity $\kappa_L = \kappa - \kappa_e$ is shown in Fig. 3b. $\kappa_L$ decreases dramatically with the increasing Ru content. As shown in Fig. 3b, the "crystalline" peak in the $\kappa_L(T)$ curve becomes more and more indistinct with the increasing Ru

content, indicating that the filling of more Ru atoms into the vacant sites results in higher lattice disorder. For the samples with $x = 0.2$ and 0.5, the $\kappa_L$ vs. $T$ curve is the classical dome shape with a "crystalline" peak around 50 K which is similar to those of typical Heusler materials such as ZrNiSn[15] and TiCo$_2$Sn[22]. The lattice thermal conductivity of the sample with $x = 0.8$ increases monotonously with the temperature, showing a glass-like thermal transport behavior similar to the amorphous SiO$_2$[16]. The unusually low lattice thermal conductivities of TiRu$_{1+x}$Sb samples can be ascribed to the increased structural disorder due to the partial filling of Ru in two sites, which is also reflected by their low sound velocities (Fig. 3d) and large crystal anharmonicities (see Supplementary Section 2 and Table S3). Noticeably, the lattice thermal conductivities of TiRu$_{1+x}$Sb samples are much lower than the typical HH compounds (Fig. 3c) and are comparable to those state-of-the-art thermoelectric materials, such as IV–VI compounds[23,24], Ge–Te alloys[25], filled skutterudites[26], etc. The HH-like TiFe$_{1.33}$Sb sample shows also pretty low total and lattice thermal conductivies[6], further confirming that the filling of vacant 4d site of the HH compound can effectively reduce the thermal transport.

The electrical transport properties of the HH-like TiRu$_{1+x}$Sb materials should be more attractive because the continuously adjustable Ru compositions could change the electronic band structure. Very interesting, TiRu$_{1+x}$Sb shows a tunable semiconducting behavior within a wide composition range, covering both p- and n-type materials (see Supplementary Fig. S4 for the electrical transport properties of all TiRu$_{1+x}$Sb samples). The Seebeck coefficient changes from positive (p-type) to negative (n-type) value with the increasing amount of Ru. Figure 4a shows the temperature-dependent Seebeck coefficient of TiRu$_{1+x}$Sb. The samples with $x < 0.3$ have positive Seebeck coefficients, and a value as high as 202 $\mu$V/K at 520 K is achieved in the sample with $x = 0.25$. These materials thus show the clear potential as

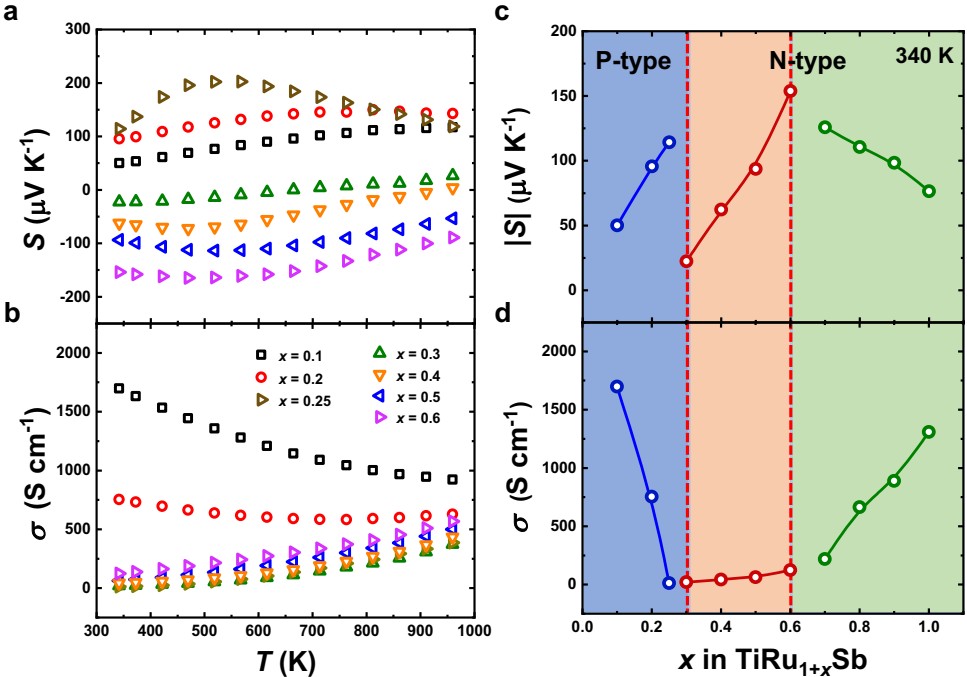

**Fig. 4 Electrical transport properties of TiRu$_{1+x}$Sb samples. a, b** Temperature-dependent Seebeck coefficients (**a**) and electrical conductivities (**b**) above room temperature. **c, d** Ru content-dependent Seebeck coefficients (**c**) and electrical conductivities (**d**) at 340 K.

promising p-type thermoelectric semiconductors. The p-type samples show typical degenerated semiconducting behavior since their electrical conductivities decrease with the temperature. The samples with $x > 0.3$ display n-type conduction with negative Seebeck coefficients (see also Supplementary Fig. S4). The Seebeck coefficient of the sample with $x = 0.6$ reaches $-165\ \mu$V/K at 470 K, suggesting the promising potential as an n-type thermoelectric material. In parallel with the enhancement of Seebeck coefficients, the electrical conductivities for those n-type samples also increase with the temperature, displaying non-degenerate or intrinsic semiconducting behavior. The TiRu$_{1.3}$Sb sample, with the composition between p- and n-type samples, shows nearly an intrinsic semiconducting behavior since it has not only the lowest Seebeck coefficient (absolute value) but also the lowest electrical conductivity among all the samples. The temperature-dependent electrical properties of the TiRu$_{1.3}$Sb sample, i.e., the electrical conductivity increases while the Seebeck coefficient changes from negative to positive value with increasing temperature, also agree with the characteristic of an intrinsic semiconductor.

To further elucidate the composition-dependent electrical transport properties of the TiRu$_{1+x}$Sb samples, the Seebeck coefficients and electrical conductivities of those samples at 340 K are respectively plotted in Fig. 4c, d. In the p-type composition region ($x < 0.3$), the Seebeck coefficient increases significantly but the electrical conductivity decreases essentially with increasing Ru content (Fig. 4a). The intuitional explanation may be that filling Ru could increase the number of minority carriers (electrons) and, thus, reduce the hole concentration. As discussed later, the picture is even more complex because adding Ru revises the band structure and tunes carrier concentration simultaneously. As expected, n-type semiconducting is achieved by further increasing the Ru content to $x \geq 0.3$ because the filling of Ru into the vacant site donates extra electrons to the matrix. In the composition region with $0.3 \leq x \leq 0.6$, both the absolute values of the Seebeck coefficient and electrical conductivity increase with the Ru content, in contrast to the transport behavior of a normal heavily doped semiconductor. This is due to the fact that electrons

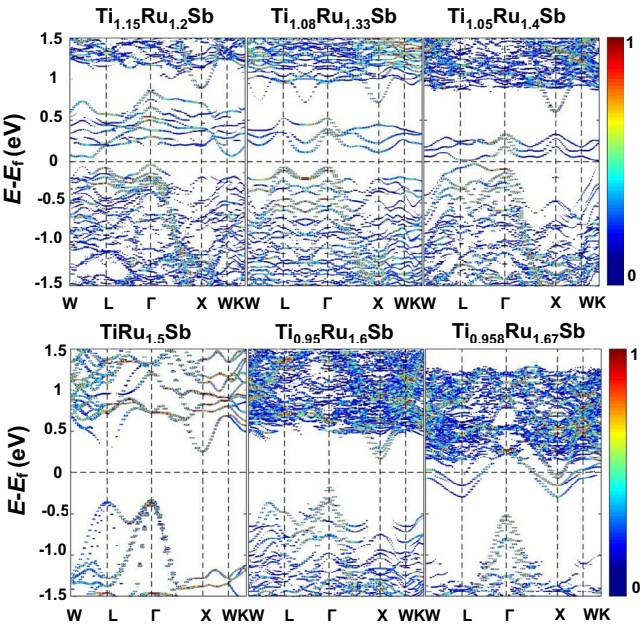

**Fig. 5 Electronic band structures of TiRu$_{1+x}$Sb materials.** The color scale represents the band weight at $(\vec{k_i}; \varepsilon_j)$.

gradually become the dominant majority carriers, consistent with the picture that the filled Ru donates electrons to the TiRu$_{1+x}$Sb sample. This results in an interesting transport transition area with mixing p-type and n-type carriers and parallel increasing of Seebeck coefficient with electrical conductivity. For $x > 0.6$, the absolute value of the Seebeck coefficient decreases while the electrical conductivity increases monotonously as usual for the heavily doped n-type semiconductors, which can be mainly ascribed to the shift of Fermi level toward the conduction band (see Fig. 5). Thus, the change of the semiconducting behavior

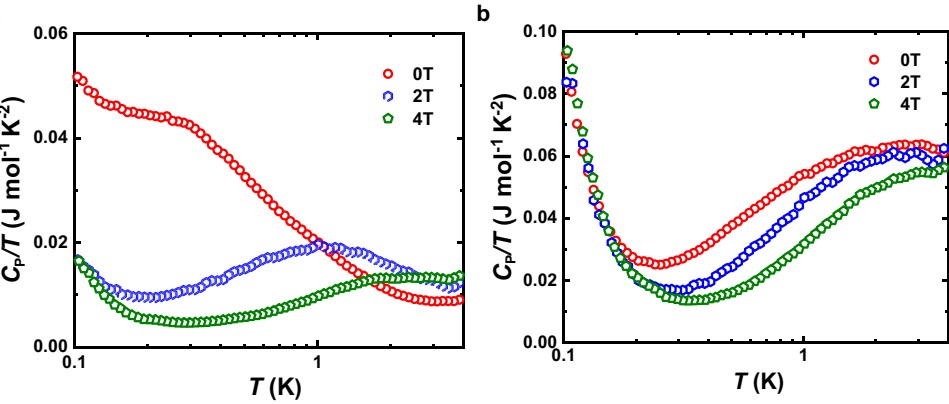

**Fig. 6 Low-temperature specific heat capacities of TiRu$_{1+x}$Sb. a** TiRu$_{1.2}$Sb. **b** TiRu$_{1.5}$Sb.

with the continuous filling of Ru can be clearly indicated by the composition-dependent electrical transport properties shown in Fig. 4c, d.

The above analysis on electrical transport properties demonstrates that both p- and n-type semiconductors can be achieved in the TiRu$_{1+x}$Sb samples by controlling the Ru content. It is quite unusual that TiRu$_{1+x}$Sb shows semiconducting behavior in such a wide composition range, which obviously violates the valence electron rules for Heusler semiconductors. In addition, both the electrical conductivity and Seebeck coefficient (absolute value) increase with the increasing Ru content for the n-type samples with $x = 0.3–0.6$, which is also curious. These extraordinary phenomena suggest that the filling of Ru not only changes the carrier concentration but also modifies the electronic band structure of TiRu$_{1+x}$Sb.

To clarify this, the first-principles-based cluster-expansion approach is used to search for the stable crystal structures of TiRu$_{1+x}$Sb compounds for different compositions, and the electronic band structure of a typical crystal structure has been calculated (Fig. 5). The calculations reveal two different regions showing different characteristics of band structures, with the composition TiRu$_{1.5}$Sb as the dividing line. TiRu$_{1.5}$Sb is the most stable composition obtained from the cluster expansion. In the TiRu$_{1.5}$Sb sample, the HH TiRuSb and FH TiRu$_2$Sb blocks stack alternatively to form the layered structure with the space group $R3m$. The TiRu$_{1.5}$Sb sample with 21 valence electrons is a Slater-Pauling semiconductor, which is very similar to the previously reported TiFe$_{1.5}$Sb[27]. The samples with $x \geq 0.5$ have a semiconductor-like band structure with a clean gap area, implying typical semiconductors with heavy n-type doping. Band structure analysis proves that the $4d$ orbitals of Ru are not completely occupied, indicating some Ru atoms lose electrons as an electron-donating impurity. The valence band maximum (VBM) of TiRu$_{1.5}$Sb locates at $\Gamma$ point and its conduction band minimum (CBM) resides at X point (see the electronic band structure of TiRu$_{1.5}$Sb in Fig. 5, and see also Supplementary Table S4 for the calculated VBM and CBM effective masses), which are mainly formed by Ti at the 4a site and Ru at the 4c/4d site (see Supplementary Fig. S5). With the increase of Ru content ($x > 0.5$), the VBM and CBM approach each other and the Fermi level shifts upward to the conduction band, but a bandgap between the CBM and VBM still exists (see the bottom panel of Fig. 5). For TiRu$_{1+x}$Sb compositions with $x < 0.5$, there are localized states within the gap area, and a small gap also exists between those localized states with normal valence band edges. Inspection based on calculations reveals that those states mainly come from the $t_{2g}$ orbital of transition metals at tetrahedral interstitial sites, including Ru, and a tiny amount of Ti migrated

to the vacant sites at low Ru contents. The transition metals at the tetrahedral interstitial sites lose only part of their $d$ electrons and meanwhile induce a series of defect states within the gap area according to the density of states (see the upper panel of Fig. 5 and Supplementary Fig. S6). The ideally ordered TiRu$_{1.5}$Sb is a semiconductor with an indirect bandgap of ~0.6 eV (see Supplementary Fig. S5), but the gap shows fluctuation to some extent due to structural disorder and composition deviation. Our calculations indicate that the separation of p-type from n-type semiconductors happens between $x = 0.4$ and $x = 0.5$. Even if the calculated composition showing conduction-type change is a little different from the experiment, the general trend is clearly visible and consistent with experimental observations. The uncertainty may come from large structure diversity as well as from the well-known DFT-method deficiency. Nevertheless, the different band structure characteristics for TiRu$_{1+x}$Sb at low and high Ru contents are well clarified.

The above understanding implies that the semiconducting behavior of TiRu$_{1+x}$Sb could be strongly correlated to the $d$ electrons of Ti and Ru, which should be especially true for the TiRu$_{1+x}$Sb materials with $0 < x < 0.5$. To further clarify this point, low-temperature specific heat capacities have been measured under different magnetic fields. For the sample with $x < 0.5$, the temperature-dependent specific heat shows a broad convex peak on the $C_P/T$ vs. $T$ curve, which shifts to higher temperature with increasing magnetic field (Fig. 6a and Supplementary Fig. S7). This phenomenon disappears for the sample with $x \geq 0.5$ (Fig. 6b and Supplementary Fig. S7). The presence of the broad convex peak, as well as its movement with the magnetic field, are closely correlated to the energy splitting of the $d$ electron orbit near the Fermi level, which can be depicted by the Schottky term of the specific heat capacity[28,29]. The external magnetic field increases the energy splitting of $d$ orbitals and drives the Schottky-term peak of the specific heat capacity to high temperature. Careful analysis on the specific capacity of TiRu$_{1+x}$Sb shows that a Schottky term does exist for the TiRu$_{1.2}$Sb sample while it is absent for the TiRu$_{1.5}$Sb sample (See Supplementary Section 3, Fig. S8 and Table S5). Hence, it is the Schottky term, associated with the $d$ orbitals of transition metal at the tetrahedral interstitial sites, that leads to the change of the specific heat capacity with the applied magnetic field. In brief, the measured low-temperature specific heat capacity obviously supports the picture of different effects of $d$ electrons on the band structures and electrical transport properties, which agrees well with the theoretical calculations. Actually, the upturn of electronic specific heat as the temperature approaching 0.1 K or lower also hints at $d$-related electronic Fermi surface states formed in TiRu$_{1.2}$Sb and TiRu$_{1.5}$Sb, which is worth studying in the future.

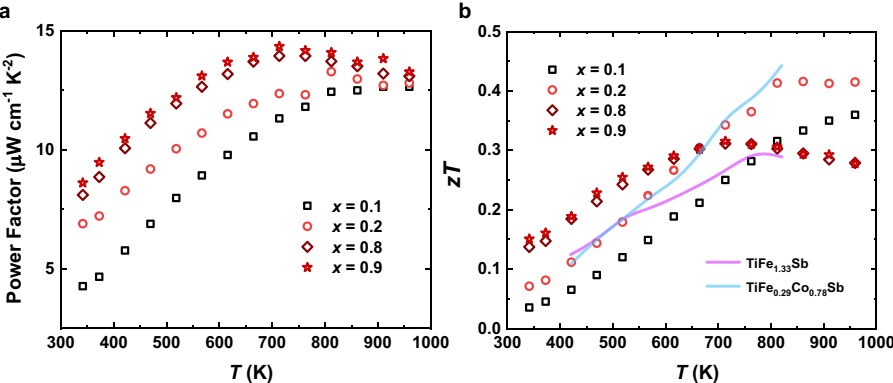

**Fig. 7 Thermoelectric properties of TiRu$_{1+x}$Sb. a** Thermoelectric power factors. **b** zT values. The temperature-dependent zT values of p-type TiFe$_{1.33}$Sb[6] and TiFe$_{0.29}$Co$_{0.78}$Sb[6] are included for comparison.

Owing to the continuously tunable compositions and electronic band structures, competitive p- and n-type electrical transport properties including outstanding Seebeck coefficients and decent electrical conductivities are realized in the Ru-filled TiRu$_{1+x}$Sb samples (see Supplementary Fig. S9 for the thermoelectric properties of all TiRu$_{1+x}$Sb samples). Thermoelectric power factors of 13.3 and 14.0 μW cm$^{-1}$ K$^{-2}$ are achieved for the p-type TiRu$_{1.2}$Sb sample at 811 K and the n-type TiRu$_{1.8}$Sb sample at 710 K (Fig. 7a), respectively. In combination with the extremely low thermal conductivities originating from high lattice disorder, promising thermoelectric properties are realized in the HH-like TiRu$_{1+x}$Sb materials. Appreciable zT (dimensionless thermoelectric figure of merit) values ~0.4 at 811 K and ~0.3 at 710 K are obtained for the p-type TiRu$_{1.2}$Sb and n-type TiRu$_{1.8}$Sb (Fig. 7b), respectively. In comparison with typical HH thermoelectric semiconductors (see Supplementary Fig. S10), the power factors and zT values of our TiRu$_{1+x}$Sb samples are much lower. Meanwhile, the zT value of our p-type TiRu$_{1.2}$Sb sample is nearly identical to that of the p-type TiFe$_{0.29}$Co$_{0.78}$Sb while much higher than that of the p-type TiFe$_{1.33}$Sb[6], which also implies that the thermoelectric properties of the TiRu$_{1+x}$Sb samples can be further improved by doping, substituting, and so on.

In summary, we demonstrate that fascinating HH-like materials with intriguing physical properties can be realized by manipulating the occupancy of the tetrahedral interstitial sites of the HH structure with appropriate elements, which bridges the HH–FH composition. The solid solution TiRu$_{1+x}$Sb with continuously tunable composition ($0.15 < x < 1.0$) and adjustable p- to n-type semiconducting thermoelectrics is discovered by filling Ru to the vacant sites. By adjusting the Ru content, the TiRu$_{1+x}$Sb samples change from p-type conduction at low Ru content ($x < 0.30$) to n-type conduction at high Ru content ($x > 0.30$) because the additional Ru behaves as an electron donor. The electronic band structure and carrier concentration/type of TiRu$_{1+x}$Sb vary with the sample composition simultaneously, leading to the abnormal semiconducting behaviors of the samples with $0.15 < x < 0.5$. In addition, the TiRu$_{1+x}$Sb samples have much lower lattice thermal conductivities than the conventional HH materials due to the structural and chemical bonding disorder breaking the transitional symmetry in the material. The sample with high Ru content shows even a glass-like thermal transport behavior. Our strategy to expand the Heusler family demonstrates a direction to explore fascinating functional materials through bridging two analog families using structure engineering. The discovery of Ru-based HH-like compounds may offer a broad prospect for exploring Heusler compounds with various exciting properties.

## Methods

**Sample synthesis**. Polycrystalline samples with nominal compositions of TiRu$_{1+x}$Sb ($x = 0, 0.1, 0.2, 0.3, 0.4, 0.5, 0.6, 0.7, 0.8, 0.9,$ and 1.0) were prepared by a combination of arc-melting, high-energy ball milling, and spark plasma sintering (SPS). The starting materials (Ti shots 99.6%, Ru pieces 99.9%, and Sb shots 99.999%) weighed according to the nominal composition were arc-melted in a high-purity argon atmosphere for 5 times to ensure homogeneity. The obtained ingots were then loaded into the Ar-protected stainless-steel jar and ball-milled for 5 h using the SPEX 8000M Mixer/Mill. Subsequently, the ball-milled powders were placed into a graphite die with an inner diameter of 12.7 mm and consolidated into pellets under 50 MPa by SPS at 1173 K for 5 min (see Supplementary Table S2 for the theoretical, measured, and relative densities of the samples).

**Composition and crystal structure characterization**. The chemical compositions of the samples were measured by an EPMA (8050G, SHIMADZU, Japan). Phase identification and crystal structure analysis were carried out with high-resolution powder XRD patterns collected from the X-ray diffraction beamline (BL14B1) at Shanghai Synchrotron Radiation Facility. Rietveld refinement of the SR-XRD pattern was performed using the WinCSD software[30]. The microstructures of the samples were examined by a high-resolution TEM (JEM-F200, JEOL, Japan) and a probe Cs-corrected TEM (Themis ETEM, Thermo Fisher Scientific, USA). The TEM specimens were prepared by mechanical slicing, polishing, and dimpling, followed by ion-milling with liquid nitrogen. The dark field TEM image and HAADF imaging were performed in the STEM model. Energy-dispersive X-ray spectroscopy was used to determine the elemental distribution at the nanoscale.

**Transport properties measurements**. High-temperature electrical transport properties and low-temperature thermoelectric properties were performed on bar-shaped samples, which were directly cut from the SPS pellet. High-temperature electrical conductivity ($\sigma$) and Seebeck coefficient ($S$) were measured by a four-probe method using the ZEM-3 system (ULVAC-RIKO, Japan). Low-temperature thermoelectric transport properties (5–350 K) including $\sigma$, $S$, and $\kappa$ were simultaneously measured via the thermal transport options using a physical properties measurement system (PPMS, Quantum Design, USA). High-temperature thermal conductivities were calculated by $\kappa = \lambda \rho C_P$, where $\lambda$ is the thermal diffusivity, $\rho$ is the density of the sample and $C_P$ is the specific heat capacity. $\lambda$ was measured by the laser flash method (LFA 467, NETZSCH, Germany), $C_P$ is estimated from the Dulong-Petit law, and $\rho$ was measured by the Archimedes method. The room-temperature sound velocity was measured by the ultrasonic material characterization system (UMS-100, TECLAB, France). Low-temperature specific heat capacities were measured by the PPMS with a dilution refrigerator option (PPMS-DR, Quantum Design, USA).

**Theoretical calculations**. All the first-principles calculations were performed by the VASP code[31] with Strongly Constrained and Appropriately Normed Semilocal Density Functional[32] and PAW potentials[33]. The TiRu$_{1.5}$Sb structure was found by the structure search through the cluster expansion included in the Alloy Theoretic Automated Toolkit[34]. During the structure search, the cut-off energy was set to 400 eV and the number of k-Points Per Reciprocal Atom was set to 1000. After finding the most stable structure of TiRu$_x$Sb for each value of $x$, a supercell was constructed, and the occupancy ratio of Ru at the 4c site to Ru at the 4d site was adjusted a little to search for more stable Ti$_{1+y}$Ru$_x$Sb structures and also made the ratio consistent with experimental observation and carrier types. The band structures of all Ti$_{1+y}$Ru$_x$Sb were unfolded into the Brillouin zone of the FCC unit cell using the BandUP code[35].

**Reporting summary**. Further information on research design is available in the Nature Research Reporting Summary linked to this article.

## Data availability

We declare that all other data supporting the findings of this study are available within the article and Supplementary Information files, and are also available from the corresponding authors upon reasonable request. Source data are provided with this paper.

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

## Acknowledgements

This work was supported by the National Key Research and Development Program of China (Nos. 2018YFA0702100, 2019YFA0704901, and 2018YFB0703600) and the National Natural Science Foundation of China (Grant nos. 51632005 and 51772186). W.Z. also acknowledges the support from the Guangdong Innovation Research Team Project (No. 2017ZT07C062), Guangdong Provincial Key-Lab program (No. 2019B030301001), Shenzhen Municipal Key-Lab program (ZDSYS20190902092905285), and the Centers for Mechanical Engineering Research and Education at MIT and Southern University of Science and Technology, China. Computing resources were supported by the Center for Computational Science and Engineering at Southern University of Science and Technology. The authors acknowledge beamline BL14B1 at Shanghai Synchrotron Radiation Facility for providing the beam time. We thank Dr. DC Wu at Thermo Fisher Scientific Company for assistance in performing atom-resolved EDS maps.

## Author contributions

J.L. conceived and designed the study. Z.-R.D. prepared the samples. Y.G., C.-Y.W., and K.G. analyzed the crystal structure. Y.J. and Z.-Y.Y. performed the microstructure analysis on TEM. Z.-R.D., J.-Y.Z. and J.L. measured and analyzed the electrical and thermal transport properties. Z.-R.D., W.-Q.Z., and J.L. measured and analyzed the low-temperature specific heat capacities. S.-H.T. and Y.-B.Z. performed theoretical calculations. Z.-R.D., J.L., and W.-Q.Z. analyzed the experimental results systemically and co-wrote the paper.

## Competing interests

The authors declare no competing interests.
