## [Peer review file · Nature Communications]

REVIEWER COMMENTS

Reviewer #1 (Remarks to the Author):

NCOMMS-21_37469_T

Half-Heusler-like compounds with wide continuous compositions and tuneable p- to n-type semiconducting thermoelectrics, by Z. Dong, C. Wang, Y. Jiang, S. Tan, Y. Zhang, Y. Grin, Z. Yu, K. Guo, J. Zhang and W. Zhang is an interesting paper, dealing with materials "between" full and half Heuslers, p- and n-type, however, this manuscript is not outstanding enough to be published in a high-ranking journal as Nature Communications and should be transferred to a lower ranking (impact factor <10) journal. As reader one expects that any article published in Nature Communications presents either something new, which is scientifically important, or it shows outstanding results. Both is not the case in the current manuscript. Compounds tunable between p- and n-type are common among thermoelectric materials since years, not only for Heuslers and half Heuslers but also for skutterudites, clathrates...The figure of merit with $ZT \sim 0.3$ and $ZT \sim 0.4$ is very poor and does not show an improvement in comparison to the cited work of Tavassoli et al. The outstanding low lattice thermal conductivity, similar to the one in Tavassoli et al., does not help to balance out the very low electrical conductivity, together with an average Seebeck coefficient it does not help to enhance ZT.

Besides that, even for a publication in an other journal some improvements are necessary:

.) The English needs improvement, therefore a native English-speaking person should correct the text. Some (out of many) mistakes:

.) A sentence should never start with "however"!

.) Problems with articles e.g.

.....no diffraction peaks from impurity phase are detected (from an impurity phase OR from impurity phases)

The diffraction peak (220) shifts continuously toward lower 2θ angle ... (a lower..)

Similar off-center location of atoms (A similar...)

.) The κL decreases significantly with the increasing of filled Ru content (no proper English!)

And so on.

.) The authors should measure the density of each sample and calculate the relative density as densities influence physical properties like electrical or thermal conductivity.

.) Tavassoli (ref. 6) published also very thermal and lattice thermal conductivities. Please read and compare and add.

.) The highest ZT of ref. 6 should be plotted as line in the graph of ZTs for comparison.

Reviewer #2 (Remarks to the Author):

The manuscript reports on finding of high structural flexibility in half-Heusler-like compound $\text{TiRu}_{1+x}\text{Sb}$. The tetrahedral Wyckoff positions 4c and 4d are shown to be continuously filled by ruthenium in range, $x = 0.15 - 1$. Ru-rich samples exhibit one of the lowest thermal conductivities reported within Heusler family (c.a. 1.7 W/mK at proximity of room temperature). Filling of the ruthenium sites leads to transition from p- to n-type conductivity. Eventually, maximum ZT of 0.4 at 800-1000 K is reported for sample $\text{TiRu}_{1.2}\text{Sb}$.

The core part of the manuscript, i.e. report of the unusual structural properties, is based on high quality data: synchrotron radiation XRD, quantitative mapping of chemical composition (EPMA), and atomic-resolution TEM. The performed Rietveld analysis carefully considered different possible types of crystallographic disorder. The authors justify the effect of p- to n-type conductivity transition by

significant changes in the calculated band structures as a function of Ru content, and occurrence of Schottky effect in heat capacity only in one of the Ru-poor sample. The above arguments are qualitative. Attempt of more precise explanation of this important and rather surprising effect is likely to attract attention of the scientific community.

The thermoelectric properties are described briefly. One of the critical information missing in the manuscript is the value of the ab-initio effective masses, which could be useful for assessing opportunities of performance optimization e.g. via parabolic band modelling. At very least, the authors should provide value of the density of effective (DOS) effective mass for $\text{Ti}_{1.15}\text{Ru}_{1.2}\text{Sb}$ sample, for which DOS data is explicitly shown in Fig. S6; preferentially the effective mass should be shown for all the compositions, for which DOS was calculated. The manuscript contains two more findings that were only scarcely discussed, and might be of interest for detailed studies in the next articles: (1) rising anharmonicity quantified by Gruneisen parameter towards ruthenium-rich compositions, (2) breakdown of 18-valence electron scheme in half-Heusler-like phase while maintaining semiconducting transport properties. ZT of 0.4 at elevated temperatures seems promising from perspective of further optimization e.g. tuning of carrier concentration by aliovalent doping.

Overall, the manuscript is free from major flaws and provides information important for scientific communities focused on solid state crystallography and applied thermoelectricity. Hence, I recommend it for publication in Nature Communications after two more technical remarks are incorporated:

1. Beginning of 7th page

"Figure 2b shows an atom-resolved high-angle annular dark field (HAADF) image and its corresponding atomic-scale EDS maps projected along a [100] direction, which indicate that Ti and Sb atoms respectively locate at the 4a and 4b sites of the F-43m structure, and Ru atoms occupy the 4c and/or 4d site."

I understand that off-centering of Sb atoms to 16e positions might be too small to be unequivocally confirmed by TEM imaging, yet I suggest maintaining the notation of Sb atoms residing on 16e Wyckoff slot, rather than 4b. This a minuscule issue, yet consistent notation will make manuscript easier to follow for the reader.

2. 2nd paragraph on 12th page

"As discussed later, the picture is even complex because adding Ru revises the band structure and tunes carrier concentration simultaneously."

Correct version: "As discussed later, the picture is even more complex (...)"

Reviewer #3 (Remarks to the Author):

Half-Heusler compound family offers a series of promising potential semiconductors for future electronic devices. I am glad to see a comprehensive study of $\text{Ru}_{(1+x)}\text{TiSb}$ compounds.

In this work, the authors successfully fabricated the $\text{Ru}_{(1+x)}\text{TiSb}$ from $x=0.15$ to 1.0 and prove the tunable phase region from half-Heusler to full-Heusler. The extremely low total and lattice thermal conductivities is really impressive, much better than any other half-Heusler compounds so far. The temperature-dependent Seebeck coefficients of the samples show the great potential as p-/n-type thermoelectric semiconductors. First-principle calculation also explains the different semiconducting behaviour with different Ru doping, supporting the experimental results that the filling of Ru changes

the carrier concentration and band structure of compounds very well. Finally, the appreciable thermoelectric power factor at 340K suggests Ru(1+x)TiSb compounds also perform well in the electrical conductivities.

The authors offer systematic and reasonable experimental and theoretical results to support their claims. The organization of paper is clear. I strongly support the paper to be published, but I still think some minor revisions are necessary in the next step.

1. The crystal structure is not plotted well in Fig. 1. Please add a legend with fraction coordinates in unit cell to indicate the atomic sites. Replace X, Y and Z by Ru, Ti and Sb, since you only analyze one element for each atomic site.

2. The authors use single-phase Ru_{1.5}TiSb as the typical sample through the paper, and I appreciate it. I further suggest the authors cite and discuss the reference (Synthesis and characterization of Fe-Ti-Sb intermetallic compounds:

Discovery of a new Slater-Pauling phase, PHYSICAL REVIEW B 93, 104424 (2016)). The team in this reference systematically discuss the fabrication and electronics property of Fe(1+x)TiSb (x=0, 0.5, 1). Because Ru has the same valence electron numbers with Fe, I think the Ru_{1.5}TiSb and Fe_{1.5}TiSb may have the same crystal structure and behave similarly in the semiconducting. In the reference, the author claims the Fe_{1.5}TiSb is the layered structure of FeTiSb and Fe₂TiSb in R3m phase, and it has 21 electrons per formula, matching the Slater-Pauling rule that 3 electrons per atom in each spin channel. The gap of Fe_{1.5}TiSb is 0.64eV, closed to 0.614eV in Ru_{1.5}TiSb. I suggest the author add a crystal figure of Ru_{1.5}TiSb along with Fig. 5S in Supplementary to show whether it is a layered structure of Ru₂TiSb and RuTiSb. Also add some sentences about electron numbers in the main paper, comparing it to Fe_{1.5}TiSb.

3. The blue color background in band structures in Fig. 5 makes the bands unclear. There are some 'watermark' in band structure of Ru_{1.5}TiSb. Please remove the blue background and replot the figures in a clearer format.

4. In the Fig. 7, I suggest to add some power factors or zT of other typical half-Heusler semiconductors to compare and discuss.

5. Please clean out the atoms outside the unit cell in Fig. S6. It is hard to understand the crystal structure with so many redundant atoms.

Here are my suggestion. Thank you

Response to the referee

Thanks to all the referees for their insightful and constructive comments. We have seriously considered all the comments from the reviewers and carefully revised the manuscript. Our replies and changes are listed in detail by following the referees' listing. The corresponding revision is also highlighted in the text.

Reviewer #1:

Half-Heusler-like compounds with wide continuous compositions and tuneable p- to n-type semiconducting thermoelectrics, by Z. Dong, C. Wang, Y. Jiang, S. Tan, Y. Zhang, Y. Grin, Z. Yu, K. Guo, J. Zhang and W. Zhang is an interesting paper, dealing with materials “between” full and half Heuslers, p- and n-type, however, this manuscript is not outstanding enough to be published in a high-ranking journal as Nature Communications and should be transferred to a lower ranking (impact factor <10) journal. As reader one expects that any article published in Nature Communications presents either something new, which is scientifically important, or it shows outstanding results. Both is not the case in the current manuscript.

Compounds tunable between p- and n-type are common among thermoelectric materials since years, not only for Heuslers and half Heuslers but also for skutterudites, clathrates...

Reply: Thanks sincerely to the referee for ranking our work interesting because of our discovery of something unusual, that is, *novel materials “between” full and half Heuslers, p- and n-type*. Actually, reviewers 2 & 3 also caught and addressed this point of our work for bridging the composition ‘gap’.

Our work provides *the first solid proof* that the long-existing composition gap between the half-Heusler (HH) and full-Heusler (FH) compounds can be bridged by continuously filling the vacant tetrahedral interstitial 4c/4d sites in a series compounds $\text{TiRu}_{1+x}\text{Sb}$ with appropriate atoms, which has never been known and realized before. The HH-like $\text{TiRu}_{1+x}\text{Sb}$ covers a surprisingly wide composition range of $x = 0.15 \sim 1.0$. Previous literatures did report that the vacant 4d sites in a few HH compounds could be filled to some extent, always in a much narrow composition range, such as $\text{TiFe}_{1.33}\text{Sb}$ (*Dalton. Trans.* 2018, **47**, 879-897), $\text{ZrNi}_{1+\delta}\text{Sn}$ ($\delta = 0 \sim 0.05$) (*Nat. Rev. Mater.* 2016, **1**, 16032), and $\text{MCo}_{1.5}\text{Sn}$ ($M = \text{Ti, Zr or Hf}$) (*Mater. Today. Phys.* 2020, **15**, 100251).

It is true that different doping (or substitution or filling, etc) can tune a thermoelectric material between p-type and n-type. This is the classical approach in optimizing the performance of typical thermoelectric materials. The doping method usually only optimizes the performance of a given material at a nearly fixed band structure (rigid band approach) and a given conduction type, which also shows a limited composition/concentration range. In contrast, competitive p- and n-type thermoelectric properties can be achieved in the $\text{TiRu}_{1+x}\text{Sb}$ systems by adjusting the filled Ru content in a wide range. Different from the conventional doping, our approach simultaneously reveals novel compounds with versatile structures, revises electronic bands, and leads to n-type or p-type transports. At each given Ru composition, there is still large space to optimize the TE performance by following the classical approaches.

The $\text{TiRu}_{1+x}\text{Sb}$ system, due to the widely tunable compositions and electronic band structures, exhibits versatile and fascinating physical properties. We have mainly demonstrated the surprising thermoelectric properties of the novel $\text{TiRu}_{1+x}\text{Sb}$ systems in this work. The thermoelectric properties of the system show many interesting and usual aspects, such as the increased anharmonicity at Ru-rich compositions, violation of 18 valence electrons rule in forming HH-like semiconductors (also mentioned by the reviewer 2), and abnormal semiconducting behaviors in the composition range of $0.15 < x < 0.50$ possibly due to the localized d electrons. These are all new to HH and Heusler materials. We believe that the rich and fascinating properties of $\text{TiRu}_{1+x}\text{Sb}$ will attract great attention from scientists in various research fields.

The figure of merit with $ZT \sim 0.3$ and $ZT \sim 0.4$ is very poor and does not show an improvement in comparison to the cited work of Tavassoli et al. The outstanding low lattice thermal conductivity, similar to the one in Tavassoli et al., does not help to balance out the very low electrical conductivity, together with an average Seebeck coefficient it does not help to enhance ZT .

Reply: The focus of this work is to demonstrate that the natural composition gap between HH and FH can be bridged and to show the fascinating properties of the newly-found HH-like $\text{TiRu}_{1+x}\text{Sb}$ systems. We report the very low lattice thermal conductivities of the $\text{TiRu}_{1+x}\text{Sb}$ samples because the high Ru composition ($\text{TiRu}_{1.8}\text{Sb}$) shows a glass-like thermal transport properties, even though the XRD pattern of the system still shows clear diffraction peaks of crystallinity. Optimizing zT of $\text{TiRu}_{1+x}\text{Sb}$ is not the focus of the current work. In fact, at each composition, the $\text{TiRu}_{1+x}\text{Sb}$ solid solution can be treated as a single material, whose properties can be further regulated by doping, substituting, and so on for further work.

Besides that, even for a publication in an other journal some improvements are necessary:

(1) The English needs improvement, therefore a native English-speaking person should correct the text. Some (out of many) mistakes:

Reply: Thanks a lot. We have checked the manuscript carefully, corrected these mistakes mentioned above, and tried our best to polish the language. The corresponding changes have been highlighted in the text.

1) A sentence should never start with “however”!

Reply: We have revised according to your suggestion.

2) Problems with articles e.g.

.....no diffraction peaks from impurity phase are detected (from an impurity phase OR from impurity phases)

Reply: We have changed it into “.....no diffraction peaks from impurity phases.....” according to your suggestion.

The diffraction peak (220) shifts continuously toward lower 2θ angle ... (a lower..)

Reply: We have corrected according to your suggestion.

Similar off-center location of atoms (A similar...)

3) The κ_L decreases significantly with the increasing of filled Ru content (no proper English!)

And so on.

Reply: We have changed the sentence into “ κ_L decreases dramatically with the increasing Ru content.”

(2) The authors should measure the density of each sample and calculate the relative density as densities influence physical properties like electrical or thermal conductivity.

Reply: Thank the referee. We have measured and calculated the densities of the samples, which are now presented in the supporting information as **Table S5**.

Table R1 Mass densities of the $\text{TiRu}_{1+x}\text{Sb}$ samples.

Ru content	Measured density (g cm^{-3})	Theoretical density (g cm^{-3})	Relative density* (%)
$x = 0.1$	8.26	8.13	/
$x = 0.2$	8.32	8.41	98.92
$x = 0.3$	8.56	8.64	99.07
$x = 0.4$	8.79	8.90	98.76
$x = 0.5$	8.94	9.13	97.92
$x = 0.6$	9.13	9.40	97.13
$x = 0.7$	9.42	9.66	97.52
$x = 0.8$	9.53	9.89	96.36
$x = 0.9$	9.65	10.15	95.07
$x = 1.0$	9.84	10.38	94.79

* The measured mass density of the sample with $x = 0.1$ is higher than its theoretical mass density because it is a multi-phase sample. The relative densities of the sample with $x = 0.7 \sim 1.0$ are slightly lower than 98%, which could be ascribed to the presence of trace amounts of RuSb_2 impurities.

(3) Tavassoli (ref. 6) published also very thermal and lattice thermal conductivities. Please read and compare and add.

Reply: Thank the referee. We have added the total and lattice thermal conductivities reported in ref.6 into our Fig. 3a and 3c for comparison. The following discussions have also been added in the text: “The HH-like $\text{TiFe}_{1.33}\text{Sb}$ sample shows also pretty low total and lattice thermal conductivities⁶, further confirming that the filling of vacant 4d site of the HH compound can effectively reduce the thermal transport.”

Fig. R1 (a) Temperature-dependent total thermal conductivities (κ) and (b) lattice thermal conductivities ($\kappa - \kappa_e$) of the $\text{TiRu}_{1+x}\text{Sb}$ samples. $\text{TiFe}_{1.33}\text{Sb}^6$ with a relative density of 98% and typical HH and FH compounds are included for comparison.

(4) The highest zT of ref. 6 should be plotted as line in the graph of zT s for comparison.

Reply: Thank the referee. We have added the highest zT of ref. 6 into our Fig. 7b for comparison. The following discussions have also been added in the text: “In comparison with typical HH thermoelectric semiconductors (See Supplementary Fig. S10), the power factors and zT values of our $\text{TiRu}_{1+x}\text{Sb}$ samples are much lower. Meanwhile, the zT value of our p-type $\text{TiRu}_{1.2}\text{Sb}$ sample is nearly identical to that of the p-type $\text{TiFe}_{0.29}\text{Co}_{0.78}\text{Sb}$ while much higher than that of the p-type $\text{TiFe}_{1.33}\text{Sb}^6$, which also implies that the thermoelectric properties of the $\text{TiRu}_{1+x}\text{Sb}$ samples can be further improved by doping, substituting, and so on.”

Fig. R2 zT values of the $\text{TiRu}_{1+x}\text{Sb}$ samples. The temperature-dependent zT values of p-type $\text{TiFe}_{1.33}\text{Sb}^6$ and $\text{TiFe}_{0.29}\text{Co}_{0.78}\text{Sb}^6$ are included for comparison.

Reviewer #2:

The manuscript reports on finding of high structural flexibility in half-Heusler-like compound $\text{TiRu}_{1+x}\text{Sb}$. The tetrahedral Wyckoff positions 4c and 4d are shown to be continuously filled by ruthenium in range, $x = 0.15 - 1$. Ru-rich samples exhibit one of the lowest thermal conductivities reported within Heusler family (c.a. 1.7 W/mK at proximity of room temperature). Filling of the ruthenium sites leads to transition from p- to n-type conductivity. Eventually, maximum ZT of 0.4 at 800-1000 K is reported for sample $\text{TiRu}_{1.2}\text{Sb}$.

The core part of the manuscript, i.e. report of the unusual structural properties, is based on high quality data: synchrotron radiation XRD, quantitative mapping of chemical composition (EPMA), and atomic-resolution TEM. The performed Rietveld analysis carefully considered different possible types of crystallographic disorder. The authors justify the effect of p- to n-type conductivity transition by significant changes in the calculated band structures as a function of Ru content, and occurrence of Schottky effect in heat capacity only in one of the Ru-poor sample. The above arguments are qualitative. Attempt of more precise explanation of this important and rather surprising effect is likely to attract attention of the scientific community.

The thermoelectric properties are described briefly. One of the critical information missing in the manuscript is the value of the ab-initio effective masses, which could be useful for assessing opportunities of performance optimization e.g. via parabolic band modelling. At very least, the authors should provide value of the density of effective (DOS) effective mass for $\text{Ti}_{1.15}\text{Ru}_{1.2}\text{Sb}$ sample, for which DOS data is explicitly shown in Fig. S6; preferentially the effective mass should be shown for all the compositions, for which DOS was calculated. The manuscript contains two more findings that were only scarcely discussed, and might be of interest for detailed studies in the next articles: (1) rising anharmonicity quantified by Gruneisen parameter towards ruthenium-rich compositions, (2) breakdown of 18-valence electron scheme in half-Heusler-like phase while maintaining semiconducting transport properties. ZT of 0.4 at elevated temperatures seems promising from perspective of further optimization e.g. tuning of carrier concentration by aliovalent doping.

Overall, the manuscript is free from major flaws and provides information important for scientific communities focused on solid state crystallography and applied thermoelectricity. Hence, I recommend it for publication in Nature Communications after two more technical remarks are incorporated:

Reply: Thank the referee for the positive and insightful comments. We agree with the referee that many interest aspects of the $\text{TiRu}_{1+x}\text{Sb}$ system deserve to be further studied, and we are moving on in this direction. According to the suggestion of the referee, we have added the ab-initio effective masses in the supporting information as **Table S4**.

Table R2 Calculated VBM and CBM effective masses of the $\text{TiRu}_{1+x}\text{Sb}$ samples.

Sample	VBM m^* (m_0)	CBM m^* (m_0)
$\text{Ti}_{1.15}\text{Ru}_{1.2}\text{Sb}$	4.45	6.62
$\text{Ti}_{1.08}\text{Ru}_{1.33}\text{Sb}$	3.80	6.44
$\text{TiRu}_{1.5}\text{Sb}$	2.05	1.80
$\text{Ti}_{0.95}\text{Ru}_{1.6}\text{Sb}$	2.19	1.89
$\text{Ti}_{0.958}\text{Ru}_{1.67}\text{Sb}$	2.05	1.80

1. Beginning of 7th page

“Figure 2b shows an atom-resolved high-angle annular dark field (HAADF) image and its corresponding atomic-scale EDS maps projected along a [100] direction, which indicate that Ti and Sb atoms respectively locate at the 4a and 4b sites of the F-43m structure, and Ru atoms occupy the 4c and/or 4d site.”

I understand that off-centering of Sb atoms to 16e positions might be too small to be unequivocally confirmed by TEM imaging, yet I suggest maintaining the notation of Sb atoms residing on 16e Wyckoff slot, rather than 4b. This a minuscule issue, yet consistent notation will make manuscript easier to follow for the reader.

Reply: Thank the referee. We have made the change accordingly.

2. 2nd paragraph on 12th page

“As discussed later, the picture is even complex because adding Ru revises the band structure and tunes carrier concentration simultaneously.”

Correct version: "As discussed later, the picture is even more complex (...)"

Reply: Thanks a lot for pointing out our language mistake. We have checked the manuscript carefully, corrected these mistakes mentioned above, and polished the language.

Reviewer #3:

Half-Heulser compound family offers a series of promising potential semiconductors for future electronic devices. I am glad to see a comprehensive study of $\text{Ru}_{1+x}\text{TiSb}$ compounds.

In this work, the authors successfully fabricated the $\text{Ru}_{1+x}\text{TiSb}$ from $x=0.15$ to 1.0 and prove the tunable phase region from half-Heusler to full-Heusler. The extremely low total and lattice thermal conductivities is really impressive, much better than any other half-Heulser compounds so far. The temperature-dependent Seebeck coefficients of the samples show the great potential as p-/n-type thermoelectric semiconductors. First-principle calculation also explains the different semiconducting behaviour with different Ru doping, supporting the experimental results that the filling of Ru changes the carrier concentration and band structure of compounds very well. Finally, the appreciable thermoelectric power factor at 340K suggests $\text{Ru}_{1+x}\text{TiSb}$ compounds also perform well in the electrical conductivities.

The authors offer systematic and reasonable experimental and theoretical results to support their claims. The organization of paper is clear. I strongly support the paper to be published, but I still think some minor revisions are necessary in the next step.

Reply: Thank the referee for the positive comments.

1. The crystal structure is not plotted well in Fig. 1. Please add a legend with fraction coordinates in unit cell to indicate the atomic sites. Replace X, Y and Z by Ru, Ti and Sb, since you only analyze one element for each atomic site.

Reply: Thank the referee. We have made the change accordingly in our revised version.

Fig. R3 Schematic crystal structure of Heusler compounds.

2. The authors use single-phase Ru_{1.5}TiSb as the typical sample through the paper, and I appreciate it. I further suggest the authors cite and discuss the reference (Synthesis and characterization of Fe-Ti-Sb intermetallic compounds: Discovery of a new Slater-Pauling phase, PHYSICAL REVIEW B 93, 104424 (2016)). The team in this reference systematically discuss the fabrication and electronics property of Fe(1+x)TiSb (x=0, 0.5, 1). Because Ru has the same valence electron numbers with Fe, I think the Ru_{1.5}TiSb and Fe_{1.5}TiSb may have the same crystal structure and behave similarly in the semiconducting. In the reference, the author claims the Fe_{1.5}TiSb is the layered structure of FeTiSb and Fe₂TiSb in R3m phase, and it has 21 electrons per formula, matching the Slater-Pauling rule that 3 electrons per atom in each spin channel. The gap of Fe_{1.5}TiSb is 0.64eV, closed to 0.614eV in Ru_{1.5}TiSb. I suggest the author add a crystal figure of Ru_{1.5}TiSb along with Fig. S5 in Supplementary to show whether it is a layered structure of Ru₂TiSb and RuTiSb. Also add some sentences about electron numbers in the main paper, comparing it to Fe_{1.5}TiSb.

Reply: Thank the referee. In our calculation, the TiRu_{1.5}Sb structure was found by the structure search through the cluster expansion included in the Alloy Theoretic Automated Toolkit (see detail information in the Methods part of our manuscript). The referee is right that the most stable TiRu_{1.5}Sb structure found by our calculation is a layered structure of TiRuSb and TiRu₂Sb with the space group R3m, which is the same as that of TiFe_{1.5}Sb reported by Naghibolashraf et al. (Phys. Rev. B 2016, 93, 104424). We have added the crystal structure of TiRu_{1.5}Sb into Fig. S5 according to your kind advice. We have also added several sentence to discuss the crystal structure and valence electrons number in the text: “The calculations reveal two different regions showing different characteristics of band structures, with the composition TiRu_{1.5}Sb as the divided line. TiRu_{1.5}Sb is the most stable composition obtained from the cluster expansion. In the TiRu_{1.5}Sb sample, the HH TiRuSb and FH TiRu₂Sb blocks stack alternatively to form the layered structure with the space group R3m. According to our theoretical calculation, the TiRu_{1.5}Sb sample with 21

valence electrons is a Slater-Pauling semiconductor, which is very similar to the previously reported $\text{TiFe}_{1.5}\text{Sb}^{25}$.”

Fig. R4. (a-f) Calculated electronic band structure and (g) crystal structure of $\text{TiRu}_{1.5}\text{Sb}$. The most stable $\text{TiRu}_{1.5}\text{Sb}$ sample has a layered structure of TiRuSb and TiRu_2Sb with the space group $R\bar{3}m$ according to our theoretical calculation.

3. The blue color background in band structures in Fig. 5 makes the bands unclear. There are some ‘watermark’ in band structure of $\text{Ru}_{1.5}\text{TiSb}$. Please remove the blue background and replot the figures in a clearer format.

Reply: Thank the referee. We have made the change accordingly.

4. In the Fig. 7, I suggest to add some power factors or zT of other typical half-Heusler semiconductors to compare and discuss.

Reply: Thank the referee. According to the suggestion of the review 1, we have added the temperature-dependent zT values of $\text{TiFe}_{1.33}\text{Sb}$ and $\text{TiFe}_{0.29}\text{Co}_{0.78}\text{Sb}$ into our Fig. 7b for comparison. In order to present the result clearly, we have presented the power factors and zT values of some typical HH semiconductors together with those of our samples in a separate figure

which is shown in the supplementary information as **Fig. S10**. The following discussions have also been added in the main text: “In comparison with typical HH thermoelectric semiconductors (See Supplementary Fig. S10), the power factors and zT values of our $\text{TiRu}_{1+x}\text{Sb}$ samples are much lower. Meanwhile, the zT value of our p-type $\text{TiRu}_{1.2}\text{Sb}$ sample is nearly identical to that of the p-type $\text{TiFe}_{0.29}\text{Co}_{0.78}\text{Sb}$ while much higher than that of the p-type $\text{TiFe}_{1.33}\text{Sb}$ ⁶, which also implies that the thermoelectric properties of the $\text{TiRu}_{1+x}\text{Sb}$ samples can be further improved by doping, substituting, and so on.”

Fig. R5 (a) Thermoelectric power factors and (b) zT values of the $\text{TiRu}_{1+x}\text{Sb}$ samples in comparison with typical HH thermoelectric semiconductors.

5. Please clean out the atoms outside the unit cell in Fig. S6. It is hard to understand the crystal structure with so many redundant atoms.

Here are my suggestion. Thank you

Reply: Thank the referee. We have revised accordingly and corresponding description about the crystal structure has also been added.

REVIEWERS' COMMENTS

Reviewer #1 (Remarks to the Author):

NCOMMS-21-37469A

Half-Heusler-like compounds with wide continuous compositions and tuneable p- to n-type semiconducting thermoelectrics, by Z. Dong, C. Wang, Y. Jiang, S. Tan, Y. Zhang, Y. Grin, Z. Yu, K. Guo, J. Zhang and W. Zhang

Although in my personal opinion this manuscript should be published in a lower ranking journal, due to the fact that both, reviewer#2 and reviewer#3, agree that NCOMMS is the proper one, I certainly give in.

In general, the authors have improved the manuscript according to the reviewers' suggestions.

.) Still, the English needs further improvement. The text cannot be published in the current version.

Some suggestions:

P11:

The Seebeck coefficient changes from positive (p-type) to negative (n-type) value with the increasing of Ru composition.

Change to: The Seebeck coefficient changes from positive (p-type) to negative (n-type) value with the increasing amount of Ru composition.

P12:

The temperature-dependent electrical properties of the $\text{TiRu}_{1.3}\text{Sb}$ sample, i.e., the electrical conductivity increases while the Seebeck coefficient changes from negative to positive value with the rising of increasing temperature, also agree with the characteristic of an intrinsic semiconductor.

P15:

...as the dividing line

...have a semiconductor-like band structures OR: have a semiconductor-like band structure

....there exists much localized states

Besides that there are a few small mistakes, the authors have overlooked:

.) In Fig. 3d, the description of the y-axis is not correct: as it is a velocity and not a volume, it should be v , not V

.) Suppl. The density should be given in gcm^{-3} (not gcm^{-1})

Reviewer #2 (Remarks to the Author):

My remarks were appropriately addressed. I recommend the manuscript for publication.

Reviewer #3 (Remarks to the Author):

The authors have revised the manuscript and supplementary materials by my suggestions and comments. Their replies are kind and satisfying. Good jobs! I agree to proceed it further to be published.

Reply to the referee

Thank all the referees for their insightful and constructive comments. We have seriously considered all the comments from the reviewers and carefully revised the manuscript. Our replies and changes are listed in detail by following the referees' listing. The corresponding revisions are also highlighted in the text.

Reviewer #1:

Half-Heusler-like compounds with wide continuous compositions and tunable p- to n-type semiconducting thermoelectrics, by Z. Dong, C. Wang, Y. Jiang, S. Tan, Y. Zhang, Y. Grin, Z. Yu, K. Guo, J. Zhang and W. Zhang.

Although in my personal opinion this manuscript should be published in a lower ranking journal, due to the fact that both, reviewer#2 and reviewer#3, agree that NCOMMS is the proper one, I certainly give in.

Reply: We greatly appreciate you for your kind help to improve our manuscript. In this revision, we have also tried our best to add some meaningful additions based on your first-round comments, which are helpful to show the novelty and importance of our work.

In general, the authors have improved the manuscript according to the reviewers' suggestions.

.) Still, the English needs further improvement. The text cannot be published in the current version.

Reply: Thank you very much. We have tried our best to polish the language with the help of an English specialist.

Some suggestions:

P11:

The Seebeck coefficient changes from positive (p-type) to negative (n-type) value with the increasing of Ru composition.

Change to: The Seebeck coefficient changes from positive (p-type) to negative (n-type) value with the increasing amount of Ru composition.

Reply: Thanks a lot for pointing out our mistakes. We have corrected the text according to your suggestion.

P12:

The temperature-dependent electrical properties of the TiRu_{1.3}Sb sample, i.e., the electrical conductivity increases while the Seebeck coefficient changes from negative to positive value with the rising of increasing temperature, also agree with the characteristic of an intrinsic semiconductor.

Reply: Thank you. We have corrected accordingly.

P15:

...as the dividing line

...have a semiconductor-like band structures OR: have a semiconductor-like band structure

...there exists much localized states

Reply: Thank you. We have made corrections according to your suggestions.

Besides that there are a few small mistakes, the authors have overlooked:

.) In Fig. 3d, the description of the y-axis is not correct: as it is a velocity and not a volume, it should be v , not V

Reply: Thank you. We have made the change accordingly.

.) Suppl. The density should be given in gcm^{-3} (not gcm^{-1})

Reply: Thank you. We have corrected this mistake accordingly.

Reviewer #2:

My remarks were appropriately addressed. I recommend the manuscript for publication.

Reply: Thank you very much for your positive comments.

Reviewer #3:

The authors have revised the manuscript and supplementary materials by my suggestions and comments. Their replies are kind and satisfying. Good jobs! I agree to proceed it further to be published.

Reply: Thank you very much for your positive comments.